# The Ceramide Synthase Subunit Lac1 Regulates Cell Growth and Size in Fission Yeast

**DOI:** 10.3390/ijms23010303

**Published:** 2021-12-28

**Authors:** Ignacio Flor-Parra, Susana Sabido-Bozo, Atsuko Ikeda, Kazuki Hanaoka, Auxiliadora Aguilera-Romero, Kouichi Funato, Manuel Muñiz, Rafael Lucena

**Affiliations:** 1Centro Andaluz de Biología del Desarrollo, Universidad Pablo de Olavide-Consejo Superior de Investigaciones Científicas-Junta de Andalucía, 41013 Seville, Spain; iflopar@upo.es; 2Department of Cell Biology, Faculty of Biology, University of Seville, 41012 Seville, Spain; ssabido@us.es (S.S.-B.); auxi@us.es (A.A.-R.); mmuniz@us.es (M.M.); 3Instituto de Biomedicina de Sevilla (IBiS), Hospital Universitario Virgen del Rocío/CSIC/Universidad de Sevilla, 41012 Seville, Spain; 4School of Applied Biological Science, Graduate School of Integrated Sciences for Life, Hiroshima University, Higashi-Hiroshima 739-8528, Japan; atsukoikeda@hiroshima-u.ac.jp (A.I.); b186822@hiroshima-u.ac.jp (K.H.); kfunato@hiroshima-u.ac.jp (K.F.)

**Keywords:** cell growth, cell size, ceramide synthase, Lac1, Lag1, fission yeast

## Abstract

Cell division produces two viable cells of a defined size. Thus, all cells require mechanisms to measure growth and trigger cell division when sufficient growth has occurred. Previous data suggest a model in which growth rate and cell size are mechanistically linked by ceramide-dependent signals in budding yeast. However, the conservation of mechanisms that govern growth control is poorly understood. In fission yeast, ceramide synthase is encoded by two genes, Lac1 and Lag1. Here, we characterize them by using a combination of genetics, microscopy, and lipid analysis. We showed that Lac1 and Lag1 co-immunoprecipitate and co-localize at the endoplasmic reticulum. However, each protein generates different species of ceramides and complex sphingolipids. We further discovered that Lac1, but not Lag1, is specifically required for proper control of cell growth and size in *Schizosaccharomyces pombe*. We propose that specific ceramide and sphingolipid species produced by Lac1 are required for normal control of cell growth and size in fission yeast.

## 1. Introduction

Growth is a common characteristic shared by all organisms. Mechanisms of growth control are responsible for generating the extraordinary diversity of cell sizes and shapes present in nature. Control of cell growth and size is relevant to cancer, because severe defects in cell size are a nearly universal feature of cancer cells, yet nothing is known about the underlying causes [1,2,3,4].

In budding yeast, growth control has recently been linked to production of ceramide lipids, which are generated from sphingolipids [5]. Sphingolipids are composed of a ceramide backbone that consists of a C18 long-chain base (LCB) bound to a fatty acid via an amide linkage. Complex sphingolipids are formed by the addition of a polar head group to a ceramide backbone (reviewed in References [6,7]). Ceramide synthesis begins at the endoplasmic reticulum (ER), where a serine palmitoyltransferase (SPT) condenses serine with a fatty acid. Further reactions yield a long-chain base (LCB), which, in yeast, are dihydrosphingosine (DHS) and phytosphingosine (PHS). In budding yeast, ceramide synthase is composed of three different subunits, namely Lac1, Lag1, and the regulatory subunit Lip1 [8,9,10]. Both Lac1 and Lag1 catalyze N-acylation of DHS or PHS to a C26 fatty acid, producing dihydroceramide (DHCer, or Cer-A) or phytoceramide (PHCer or Cer-B). Further hydroxylations of Cer-A and Cer-B at different positions can generate Cer-C or Cer-D. The polar head of ceramides can be further modified at the Golgi to generate complex sphingolipids [11]. These include inositol phosphoceramide (IPC), mannosylinositol-phosphorylceramide (MIPC), and mannosyl-diinositolphosphorylceramide (MIP_2_C). In *S. cerevisiae*, MIP_2_C is the most abundant complex sphingolipid.

While single deletions cause mild defects in budding yeast, loss of both *LAC1* and *LAG1* results in lethality [12] or severe growth defects and a drastic reduction in ceramide content [5,8,9]. Overexpression of either one or both proteins does not increase ceramide levels, thus suggesting that ceramide production is tightly controlled [9]. Recently, it has been shown that budding yeast Lag1 has an increased affinity toward PHS, while Lac1 has an affinity toward DHS [13].

Similar to budding yeast, the rod-shaped fission yeast *Schizosaccharomyces pombe* is a powerful organism in which to dissect the molecular mechanisms that govern cell growth and size control. Fission yeast grows in length by linear extension during G2 and divides at a constant cell size [14,15]. Recently, different models have been proposed to explain how cell growth and size are coordinated, but further work has to be performed to fully understand all the mechanisms involved in this fine control [16,17,18,19,20,21].

Our previous analysis indicated that ceramide signaling might play a conserved role in cell size control both in fission and budding yeast, since cells treated with myriocin, an inhibitor of sphingolipid production, cause a dose-dependent decrease in cell size and growth rate [5]. In *S. pombe*, sphingolipid metabolism and ceramide synthesis are poorly understood. Here, we characterize fission yeast ceramide synthase and discovered that specific ceramide species are required for normal control of cell growth and size. The results suggest that ceramide signaling is implicated in molecular mechanisms that coordinate cell growth and size in a variety of organisms.

## 2. Results

### 2.1. Fission Yeast Ceramide Synthases Are Conserved and Localize at the ER

The finding that ceramides are key players in control of cell growth and size in budding yeast prompted us to investigate whether the same signals are conserved in the distant related fission yeast *Schizosaccharomyces pombe*. First, we compared protein sequence conservation between fission and budding yeast ceramide synthases. Fission yeast *Schizosaccharomyces pombe* ceramide synthase is also encoded by the homologs Lac1 and Lag1. There are no Lip1 orthologs in fission yeast. Multiple sequence alignment between all the subunits shows a strong similarity (Figure 1). Fission yeast Lac1 and Lag1 shared 30.3% of identical residues (Appendix A). Interestingly, fission yeast Lac1 shares 37.7% of identical residues with Lac1 from budding yeast. In comparison, fission and budding yeast Lag1 share only 29.5% of identical residues (Appendix A). Thus, phylogenetic analysis suggests that fission yeast Lag1 belongs to a different clade (Appendix A).

In budding yeast and mammalian cells, ceramide synthase is known to localize at the endoplasmic reticulum (ER) [22,23,24]. Since we found a strong sequence conservation in fission yeast, we decided to examine whether they also share a similar localization. We fused GFP and Tomato tags to Lac1 and Lag1. Endogenous expression of Lag1-tagged protein did not show localization defects. Initial imaging of C-terminal endogenous tagging of Lac1 presented a peculiar pattern, indicating that GFP tagging at the C-terminal interfered with Lac1 localization (data not shown). To circumvent this problem, we expressed GFP-Lac1 under a thiamine-regulated *nmt1* promoter integrated at the endogenous locus. The *nmt1* promoter does not switch off completely, and this allowed us to express GFP-Lac1 adjusting the concentration of thiamine in the supplemented media. To obtain an expression similar to the endogenous level, we used media complemented with thiamine [25]. Using live-cell imaging in exponentially growing cells, we found that Lac1 and Lag1 co-localize together at a membranous structure surrounding the nuclear envelope and the plasma membrane, similar to an ER pattern (Figure 2A). To confirm that ceramide synthase subunits localize at the ER, we performed a double-labeling experiment, using the artificial luminal ER marker mCherry-AHDL [26]. In rapidly growing cells, Lac1 and Lag1 co-localized completely with the ER marker (Figure 2B), confirming that ceramide synthase localizes at the ER in fission yeast.

### 2.2. Fission Yeast Lac1 and Lag1 Co-Localize at the Endoplasmic Reticulum

In budding yeast, active purified ceramide synthase complex contains Lac1 and/or Lag1. The accessory subunit Lip1 can be found with either Lac1 or Lag1 [10]. To further characterize ceramide synthase in fission yeast, we next assessed whether Lac1 interacts with Lag1 in vivo. We analyzed the interaction by using cells expressing Lag1-3xHA and a GFP-tagged version of Lac1 under the thiamine-repressible promoter. As a control, we used single-tagged versions of each protein. Since Lac1 and Lag1 localize at the ER (Figure 2), a native co-immunoprecipitation experiment was performed on ER-enriched fractions. We used anti-GFP to precipitate GFP-Lac1, followed by Western blot analysis, using anti-HA to detect Lag1-3xHA. Immunoprecipitation in detergent-solubilized extracts revealed that Lag1-3xHA co-precipitated with GFP-Lac1 in native conditions (Figure 3A). Since Lac1 and Lag1 interacted in vivo, we next tested whether their localization at the ER is dependent on the presence of each other. Live cell imaging revealed that GFP-Lac1 and Lag1-GFP remain localized at the ER in the absence of Lag1 or Lac1 respectively, indicating that their ER localizations are not interdependent.

### 2.3. Lac1 Is Implicated in Control of Cell Growth and Size in Fission Yeast

The discovery that ceramide synthesis is required for normal control of cell growth and size in budding yeast prompted us to analyze the defects of ceramide synthase mutants in fission yeast. Double deletion of *lac1∆* and *lag1∆* was inviable in normal conditions (Appendix A Appendix A and Reference [27]). Thus, we examined defects caused by *lac1∆* or *lag1∆* single mutants. We first investigated whether ceramide synthase mutants were affected at a range of temperatures. As a result, *lac1∆*, but not *lag1∆*, showed strong defects in cell proliferation at all temperatures in a spot assay (Figure 4A). Growth defects of *lac1∆* were also observed when the cells were grown in liquid media (Figure 4B). We next sought to determine the consequences of Lac1 and Lag1 deletions at the cellular level. DAPI–Calcofluor staining of exponentially growing cells at 25 °C revealed a strong decrease in cell length at division in *lac1∆* cells (Figure 4C,D). Notably, the decrease in cell size in *lac1∆* cells was partially reduced at 36 °C. Lag1 mutants had no significant effect on cell size at division in any condition tested.

### 2.4. Lac1 and Lag1 Generate Different Species of Ceramides and Complex Sphingolipids

We next sought to determine whether the defects in cell growth and size found in *lac1∆* cells were associated with a misregulation of ceramide and sphingolipid production. It is known that yeast converts exogenous DHS, an intermediate in the synthesis of ceramide, to DHS-1P, DH-Cer or PHS, which can be further converted into PHS-1P or PH-Cer [9,28,29]. Both DH-Cer and PH-Cer are transformed into IPC-A/B’ or IPC-B/C/D, respectively (Figure 5A). To assess sphingolipid synthesis, wild-type, *lag1∆* and *lac1∆* cells were in vivo labeled with [^3^H]-DHS. The labeled lipids were extracted and separated by thin-layer chromatography (TLC). As a reference, we used wild-type and *lac1∆ lag1∆* cells from *S. cerevisiae*. Wild-type cells from *S. cerevisiae* and *S. pombe* were treated with Aureobasidin A (AbA), an inhibitor of IPC synthesis, to identify IPC species.

While no significant changes were found in Lag1-deprived cells, *lac1∆* cells exhibited major changes in lipid composition (Figure 5B). We detected an accumulation of PHS and sphingoid bases-1-phosphate levels (PHS-1P or DHS-1P). This result implies that the *lac1∆* strain might be deficient at a step that converts PHS to phytoceramides. Additionally, the pattern of complex sphingolipids in Lac1-depleted cells shows a strong accumulation of IPC and the appearance of new bands that might correspond to different IPC species.

To further characterize ceramide synthase activity, we extracted lipids from the ceramide fraction, including a specific lipid from *S. pombe* that was not present in *S. cerevisiae* (Lipid A, Figure 5B) and separated them by TLC, using a different solvent system. The pattern of lipids observed shows that *S. pombe* strains have specific species of lipids. Lipid B and C were detected in wild-type and *lac1∆* cells, but not in *lag1∆* (Figure 5C). Particularly, the lipid B level was higher in *lac1∆* compared to the wild type (Figure 5C). We found a low signal band in *lag1∆* cells that could correspond to PH-Cer (asterisk in Figure 5C).

To determine whether these specific lipids were ceramides, we extracted and treated them with NaOH, a mild base treatment that hydrolyzes glycerophospholipids. Lipids B, A and C were resistant to NaOH, confirming that they are ceramides (Figure 5D,E). To confirm the study of these ceramides we subjected them to strong hydrolysis with HCl. *S. cerevisiae* ceramide was cleaved to PHS, confirming that it is phytoceramide (PH-Cer, Figure 5D). In, *S. pombe*, lipids B, C and A were cleaved to DHS during HCl treatment (Figure 5D,E). Thus, in contrast to budding yeast, we can conclude that fission-yeast-specific ceramides are dihydroceramides (DH-Cer). Since long and short-chain ceramides run differently in the TLC, we suggest that *S. pombe* lipid A (DH-Cer1) is a short-chain DH-Cer, while lipids B (DH-Cer2) and C (DH-Cer3) are long-chain DH-Cer (see model in Figure 6).

## 3. Discussion

How cells coordinate the amount of growth required for cell division has been a fundamental question in cell biology. Recent work suggests that ceramide-dependent signals play important roles in control of cell growth and size [5]. Ceramide synthase has been studied extensively in budding yeast, but little is known about ceramide synthase in fission yeast [8,9,10,13,27,30]. Here, we present new data suggesting that the production of specific ceramide species is implicated in normal control of cell growth and size control in fission yeast. To our knowledge, this is the first functional characterization of ceramide synthase in *Schizosaccharomyces pombe*.

We found a number of similarities between fission and budding yeast that suggest the existence of conserved mechanisms. In agreement with previous data, we found that ceramide synthase subunits Lac1 and Lag1 are localized at the ER [22,23,24]. Moreover, co-immunoprecipitation experiments suggest that Lac1 and Lag1 interact in vivo. No changes in localization were detected in GFP-Lac1 *lag1∆* and vice versa, which argues that both subunits do not play an important role in each other’s localization. In budding yeast, optimal ceramide formation and localization are dependent upon phosphorylation at both the N- and C-terminus domains in Lac1 and Lag1 [24,31]. Further analysis will determine whether fission yeast ceramide synthase localization and function are also dependent on post-translational modifications. A mutagenic analysis of Lac1 to identify residues involved in localization and activity is in process and will be published elsewhere (Flor-Parra I. and Lucena R., unpublished results).

We also present new data supporting a role of ceramide signaling in control of cell growth and size in fission yeast. We discovered that loss of Lac1, but not Lag1, caused substantial reductions in growth rate and cell size. Temperature did not increase defects in proliferation, suggesting that Lac1 function is required for normal growth at all temperatures tested. Interestingly, cell size at division was partially recovered at 36 °C. Previous work showed that shifting yeast cells to high temperatures results in increased levels of ceramides [32,33], supporting the idea that ceramides might be required for normal control of cell growth and size.

Our lipid analysis also provides evidence that Lac1 and Lag1 could be responsible for generating different species of ceramides and complex sphingolipids with separate functions (see model in Figure 6). In wild-type cells, long- and short-chain PH-Cer and DH-Cer are likely converted into IPC and MIPC. Both Lag1 and Lac1 produce short DH-Cer, the substrate of a possible short chain IPC. However, Lac1 and Lag1 might have different affinities toward DHS and PHS when they synthesize long-chain ceramides. Our results suggest that Lac1 protein has a preference to produce long-chain PH-Cer that are further transformed to IPC and MIPC, while Lag1 synthesizes long-chain DH-Cer that are used to produce different species of long-chain IPC. Accordingly, *lag1∆* cells can produce PH-Cer and subsequent complex sphingolipids but are not able to synthesize long-chain DH-Cer (lipids B and C). In contrast, *lac1∆* cells do not synthesize PH-Cer but generate both short- (lipid A) and long-chain DH-Cer (lipid B and C). Increased IPC levels suggest that conversion to MIPC is affected in *lac1∆*. Since long-chain ceramides, but not short-chain ceramides, are toxic [34], the results are consistent with the normal growth of *lag1∆* and defects in *lac1∆* cells. Alternatively, the lack of PH-Cer could be responsible for the defects in cell growth and size found in *lac1∆* cells. Nevertheless, we cannot exclude other possibilities. Further work will determine which ceramide species are involved in this process.

Shui and collaborators found that *S. pombe* contains high levels of free ceramide (d18:1/18:0), short-chain PH-Cer (t18:1/20:0-B) and IPC-B (t18:1/20:0), while *S. cerevisiae* does not have these species [35]. Labeling experiments measure the de novo synthesis of ceramides and sphingolipids, whereas mass spectrometry analysis measures the level of steady-state lipids maintained by sphingolipid synthesis and catabolism. We were unable to detect PH-Cer by DHS labeling experiments. A possible explanation is that either the synthesis of detectable PH-Cer requires a longer time or the de novo synthesized PH-Cer by Lac1 is rapidly converted to IPCs. This latter conclusion might be supported, since a low-intensity band that might correspond to PH-Cer is detected in *lag1∆* cells. In these cells, labeled DHS is transformed into PHS that is further channeled to Lac1, creating an excess of PH-Cer that is not metabolized to complex sphingolipids.

The initial characterization of budding yeast ceramide synthase suggested that Lac1 and Lag1 were redundant enzymes, since single deletions caused no obvious growth defect in yeast [12,23]. Further analysis demonstrated that Lac1 and Lag1 have different affinities toward PHS and DHS to form ceramides and complex sphingolipids, and this might explain Lag1 specific role in longevity [13]. Moreover, sphingolipid synthesis in different fungi suggests that Lag1-related protein is involved in glucosyl and galactosylceramides, while Lac1-related protein is responsible for IPC and MIPC production [36,37]. Our genetic and lipid analysis strongly suggest that Lac1 is the fundamental subunit of the ceramide synthase in fission yeast. It is possible that fission yeast Lac1 carries major weight in ceramide and complex sphingolipid canonical production, while Lag1 is only necessary for specific situations. An appealing explanation is that selective ceramide signaling emanating from Lac1 is necessary for cell growth and size control, while Lag1 is implicated in longevity or stress conditions.

Overall, the results shown here reinforce our hypothesis that ceramide signaling strongly influences control of cell growth and size in eukaryotic cells. A tempting idea is that cells set the amount of growth required for cell division according to the levels of ceramides. Future analysis will aim to discover the molecular mechanisms responsible for ceramide-dependent signals that control cell growth and size.

## 4. Materials and Methods

### 4.1. Strains, Plasmids and Media

Strains used in this study are listed in Table 1. Cells were grown at 25, 30 or 37 °C, in standard YES media (MP Biomedical, Irvine, CA, USA) supplemented with 175 mg/L adenine, histidine, lysine, uracil and lysine hydrochloride (Sigma-Aldrich, St. Louis, MO, USA). Plates were made by adding 2% agar. Standard methods for *S. pombe* growth and genetics were used [38]. Strains were constructed by using a PCR-based homologous recombination method to insert markers in the yeast chromosome [39]. Constructs were checked via PCR and sequencing, and strains were outcrossed at least three times. Double mutants were generated by crosses and tetrad analysis.

### 4.2. Sequence Alignment

The identity between Lac1/Lag1 proteins in *S. pombe* and *S. cerevisiae* was calculated by using UNIPROT-Align, using the standard parameters [40] (https://www.uniprot.org/align/, accessed on 14 October 2021). Representation of the alignment was made by using ESPript 3.0 [41]. Sequences were obtained from Saccharomyces Genome Database [42] (https://www.yeastgenome.org/, accessed on 27 July 2021, Version R64-2-1) and PomBase [43] (https://www.pombase.org/, accessed on 27 July 2021, Version gky961. Cambridge, UK). A phylogenetic tree was constructed by using MEGA11 software (Version 11. www.megasoftware.net) [44].

### 4.3. Microscopy Analysis

DAPI/Calcofluor staining was performed as described in Reference [45]. Cells were visualized by using a Leica DMi8 microscope equipped with an objective lens (HCX PL APO 1003/1.40OIL PH3 CS), L5 (GFP) filter, a Hamamatsu camera and Application Suite X (LAS X) software, as described previously [46].

Cell length was measured from pictures of DAPI/Calcofluor-stained cells, using the Segmented line option of Image J (National Institute of Health, https://imagej.nih.gov/ij/, 1997–2018, Version 1.53m, Bethesda, MD, USA). Average cell length was determined from more than 250 cells, and comparison between strains was performed by using a two-tailed unpaired Student’s *t*-test. Graphical representation was performed by using GraphPad Prism version 5.00a for Macintosh, (GraphPad Software, 5.0, San Diego, CA, USA, www.graphpad.com).

For live-cell imaging, cells were typically grown in exponential phase in liquid YES medium at 25 °C with shaking for 18–24 h. In some experiments, cells were mounted in liquid YES medium directly on glass. For long-term imaging, cells were placed on microslide wells (Ibidi #80821) coated with soybean lectin (Sigma #1395). Images were generally acquired by using a spinning-disk confocal microscope (IX-81; Olympus; CoolSnap HQ2 camera (Hamburg, Germany), Plan Apochromat 100×, 1.4 NA objective; Roper Scientific). Cells were imaged in 14 z-series with a step size of 0.3 µm. Maximal projections of images were created by using Image J. A wide-field Nikon Eclipse 800 microscope with a 60 × 1.4 N.A. objective was also used for some studies. Temperature was stably controlled in the room during imaging at 25 °C, unless otherwise indicated.

### 4.4. Native Co-Immunoprecipitation

The native co-immunoprecipitation experiment was performed on enriched ER fractions, as described [46]. Briefly, 60 OD_595_ units of yeast cells were washed twice with TNE buffer (50 mM Tris-HCl (pH 7.5), 500 mM NaCl, 5 mM EDTA, 1 mM phenylmethylsulfonylfluoride and protease inhibitor cocktail; Roche Diagnostics, Basel, Switzerland) and lysed in a Mini-beadbeater 16 (BioSpec, Bartlesville, OK, USA), at top speed, for 2 min. Cell debris and glass beads were removed by centrifugation at 1000× *g* for 10 min, at 4 °C. The supernatant was then centrifuged at 13,000× *g* for 15 min, at 4 °C. The pellet was resuspended in TNE, and digitonin was added to a final concentration of 1%. The suspension was incubated for 1 h at 4 °C, with rotation, after which insoluble components were removed by centrifugation at 17,000× *g* for 60 min, at 4 °C. For immunoprecipitation of GFP-Lac1, samples were first pre-incubated with empty agarose beads (ChromoTek, Planegg, Germany) at 4 °C for 1 h and subsequently incubated with GFP-Trap_A (ChromoTek) at 4 °C for 3 h. The immunoprecipitated beads were washed five times with TNE containing 0.2% digitonin, eluted with SDS sample buffer and resolved on SDS polyacrylamide gel.

Samples were analyzed by Western blotting, as previously described [47]. Briefly, SDS–PAGE gels were run at a constant current of 20 mA, and electrophoresis was performed on gels containing 10% polyacrylamide and 0.13% bis-acrylamide. Proteins were transferred to nitrocellulose, using a Trans-Blot Turbo system (Bio-Rad, Hercules, CA, USA). Blots were separated in half and probed with primary antibody overnight, at 4 °C, on a rocker platform. Lag1-3xHA was detected by using anti-HA high affinity rat monoclonal antibody (Clone 3F10, Roche #11867423001) at a dilution of 1:5000. GFP-Lac1 was detected by using an anti-GFP rabbit polyclonal antibody (a gift from Howard Riezman, University of Geneve, Geneve, Switzerland) at a dilution 1:3000. Both antibodies were used in TBST (10 mM Tris-Cl, pH 7.5, 100 mM NaCl, and 0.1% Tween 20) containing 5% milk.

All blots were probed with an HRP-conjugated goat anti-rat secondary antibody (Thermo Scientific #31470, Waltham, MA, USA) or goat anti-rabbit (Thermo #31460) at a 1:5000 dilution in TBST. Secondary antibodies were detected via chemiluminescence with Advansta ECL reagents and a Bio-Rad ChemiDoc imaging system.

### 4.5. Lipid Analysis

In vivo labeling of sphingolipids with [^3^H] dihydrosphingosine (DHS) was carried out as described [48]. Briefly, cells grown overnight in YES media supplemented with 225 mg/L adenine, histidine, lysine, uracil and lysine hydrochloride were incubated with or without aureobasidin A (2 μg/mL) for 2 h at 25 °C, and then labeled with [^3^H] DHS at the same temperature for 2 h. Radiolabeled lipids were extracted with chloroform–methanol–water (10:10:3, *vol*/*vol*/*vol*) and analyzed by thin-layer chromatography (TLC), using solvent system I, chloroform–methanol–4.2N ammonium hydroxide (9/7/2, *v*/*v*/*v*). For ceramide analysis, the fractions containing ceramides were collected by scraping silica gels from the TLC plate and eluting the lipids from the gels with chloroform–methanol (1/1, *v*/*v*). Subsequently, ceramides were analyzed by TLC, using solvent system II, chloroform–methanol–4.2N ammonium hydroxide (40/10/1, *v*/*v*/*v*). If necessary, the extracted ceramides were subjected to strong HCl hydrolysis (1M HCl, for 1 h at 80 °C) or mild alkaline hydrolysis with NaOH (0.1 M NaOH, for 2 h at 30 °C) [11], and analyzed by TLC, using solvent system I. Radiolabeled lipids were visualized and quantified on the Typhoon FLA-7000 system (GE Healthcare, Chicago, IL, USA).

## Figures and Tables

**Figure 1 ijms-23-00303-f001:**
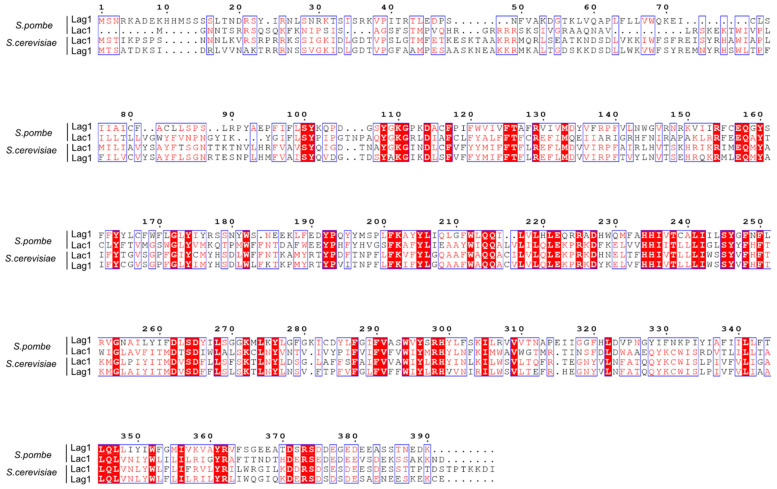
Multiple alignments of ceramide synthase subunits in fission and budding yeast. Sequence alignment was made for ceramide synthases from fission and budding yeast. White characters on red boxes denote identical residues, and red characters on white boxes denote conserved residues.

**Figure 2 ijms-23-00303-f002:**
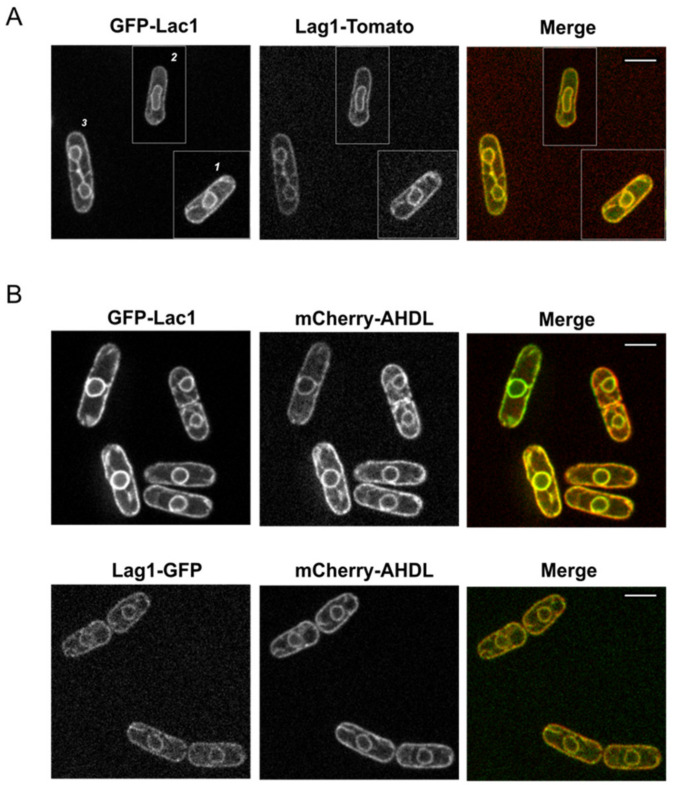
Ceramide synthase complex is localized at the endoplasmic reticulum. (**A**) Wild-type cells expressing GFP-Lac1 and Lag1-Tomato were grown to early log-phase in YES media and visualized by using live-cell imaging. Image shows a composite of three representative cells in G2 (cell 1), early mitosis (cell 2) and late mitosis (cell 3). (**B**) Representative wild-type log-phase cells expressing the endoplasmic reticulum marker mCherry-AHDL with GFP-Lac1 (upper panel) or Lag1-GFP (lower panel). Scale bar is 5 µm for all the images.

**Figure 3 ijms-23-00303-f003:**
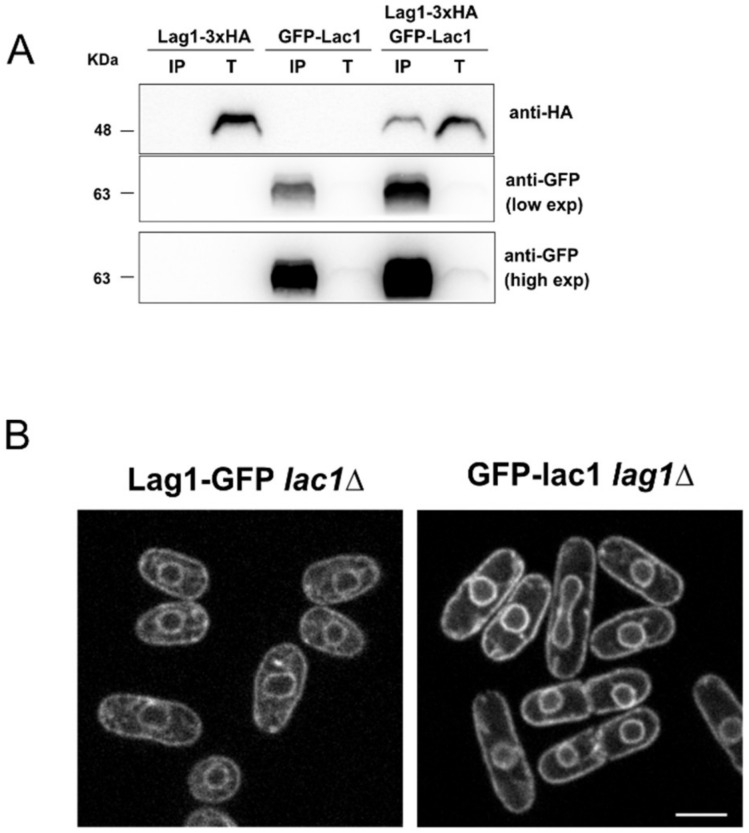
Lac1 and Lag1 co-immunoprecipitate in vivo. (**A**) Wild-type strains expressing GFP-Lac1, Lag1-3xHA or GFP-Lac1 Lag1-3xHA were grown in YES media and immunoprecipitated on ER-enriched fractions with anti-GFP antibody, followed by immunoblotting by using antisera to HA and GFP. Total (T) represents a fraction of the solubilized input material. IP represents total protein immunoprecipitated. Due to the highly efficient immunoprecipitation, image shows different exposure times (low exp/high exp) to demonstrate a band is present in the total extracts. Two biological replicates of the immunoprecipitation were processed, showing the same result. (**B**) Lag1-GFP and GFP-Lac1 localization in *lac1∆* and *lag1∆*, respectively. Medial section of spinning-disk images is shown. Scale bar is 5 µm.

**Figure 4 ijms-23-00303-f004:**
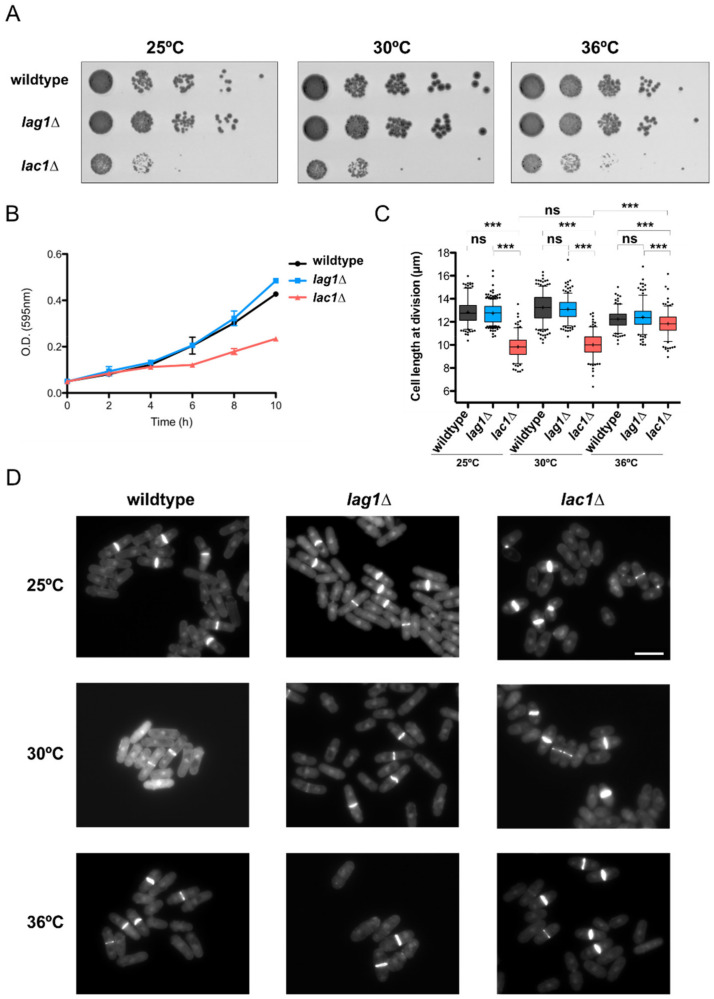
Ceramide synthase subunit Lac1 is implicated in control of cell growth and size. (**A**) A series of 10-fold dilutions of the indicated strains were grown at 25, 30 or 36 °C for 2 to 3 days on YES media. (**B**) Absorbance at 595 nm of the indicated strains grown in YES media at 25 °C. Error bar shows standard deviation from three different experiments. (**C**) Box-and-whisker plot showing cell length at division in wild-type, lag1∆ and lag1∆ strains at different temperatures. Boxes extend from the 25th to 75th percentile, while whiskers plot maximum and minimum values within the 5–95 percentiles. Points below and above whiskers show outside values. N > 250 cells for each condition; ns = not significant; *p*-value was determined by Student’s *t*-test. *** Denotes *p* < 0.0001. (**D**) DAPI–Calcofluor staining of cells grown in (**C**). Scale bar is 10 µm for all the images.

**Figure 5 ijms-23-00303-f005:**
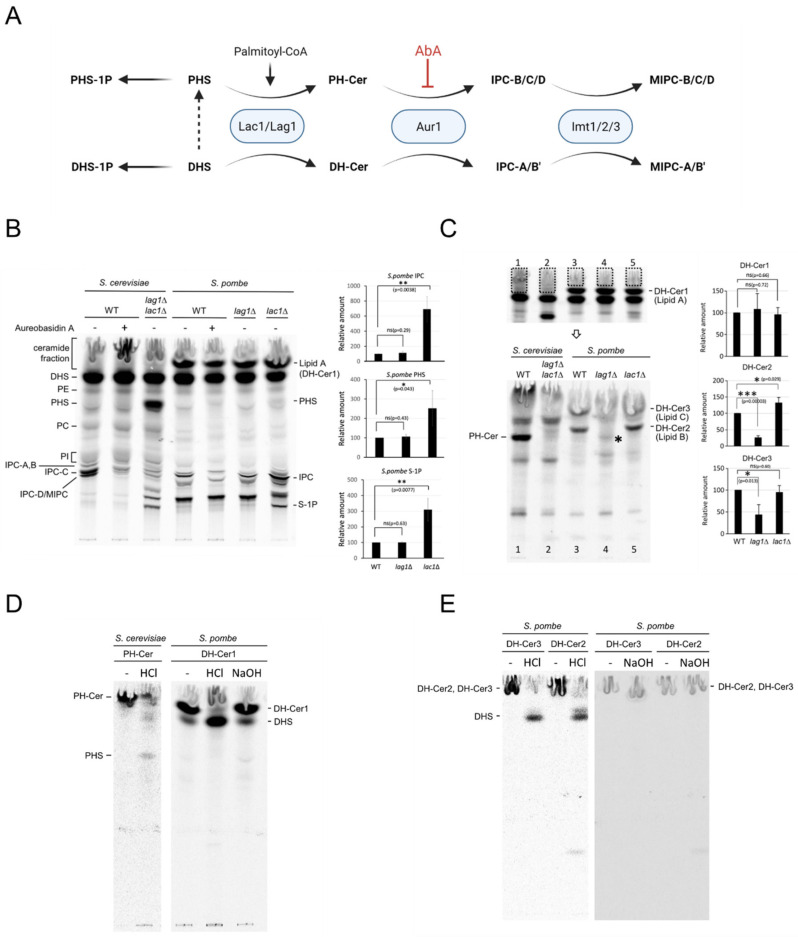
Lipid analysis in *lac1∆* and *lag1∆* cells. (**A**) Summary of sphingolipid synthesis pathway. Small-molecule inhibitor aureobasidin A is indicated in red. DHS, dihydrosphingosine; PHS, phytosphingosine; DHS-P/ PHS-P, dihydrosphingosine/phytosphingosine-1-phosphate; IPC-A, -B, -C and –D, inositolphosphorylceramide subclasses A, B, C and D; MIPC, mannosylinositolphosphorylceramide. (**B**–**D**) Wild-type and mutant cells were labeled with [^3^H] dihydrosphingosine (DHS) at 25 °C for 2 h, in the absence or presence of 2 μg/mL aureobasidin A. The labeled lipids were extracted and analyzed by thin-layer chromatography (TLC), using solvent system I ((**B**,**C**), upper). * Denotes (in (**C**)) putative PH-Cer in *S. pombe lag1∆*. Fractions containing ceramides in (**C**) upper were collected by scraping silica gels; extracted with a mixture of chloroform–methanol; and analyzed by TLC, using solvent system II (**C**, lower). Phytoceramide (PH-Cer) and dihydroceramides (DH-Cer2 and DH-Cer3) were purified from the TLC plate in C lower, and DH-Cer1 was purified from (**C**) upper. They were subjected to strong HCl hydrolysis or mild alkaline hydrolysis with NaOH and analyzed by TLC, using solvent system I (**D**,**E**). Incorporation of [^3^H] DHS into IPC (**B**) and DH-Cer (**C**) was quantified, and the relative amounts were determined as the percentage of the incorporation in wild-type cells. Data represent mean ± SD of three independent experiments; * *p* < 0.05, ** *p* < 0.01 and *** *p* < 0.001 by Student’s *t*-test. PE, phosphatidylethanolamine; PHS, phytosphingosine; PC, phosphatidylcholine; PI, phosphatidylinositol; IPC-A, -B, -C and -D, inositolphosphorylceramide subclasses A, B, C and D; MIPC, mannosylinositolphosphorylceramide; PH-Cer, DH-Cer1(lipid A), DH-Cer2 (lipid B), DH-Cer3 (lipid C), different ceramide species, S-1P, sphingoid base-1-phosphate.

**Figure 6 ijms-23-00303-f006:**
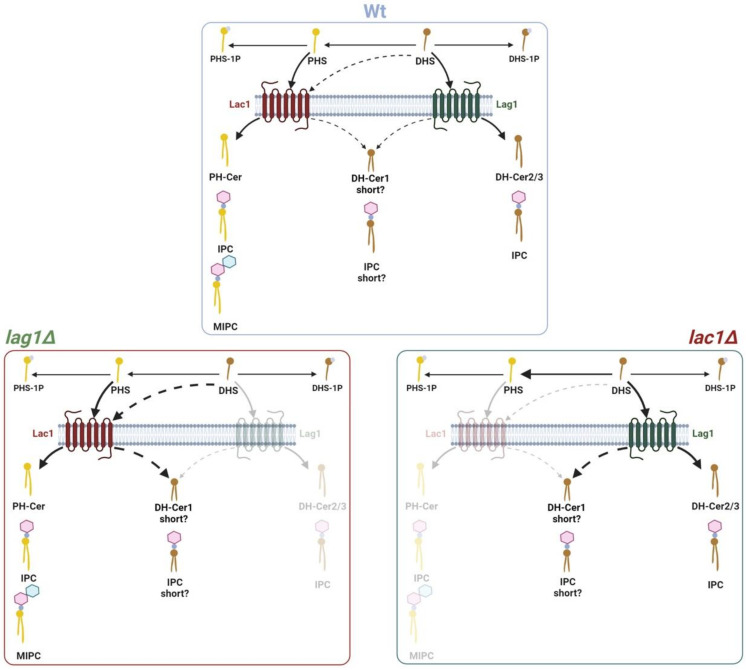
Model that explains ceramide and complex sphingolipid synthesis in wild-type, *lac1∆* and *lag1∆* cells in fission yeast. PHS, phytosphingosine; PHS-1P, phytosphingosine-1-phosphate; DHS, dihydrosphingosine; DHS-1P, dihydrosphingosine-1-phosphate IPC, inositolphosphorylceramide; MIPC, mannosylinositolphosphorylceramide; PH-Cer, DH-Cer1, DH-Cer2, DH-Cer3, different ceramide species. Figure created with Biorender.com.

**Table 1 ijms-23-00303-t001:** Strains used in this study.

Strain	Genotype	Source
NJ1	*h+ his7-366, ade6-M210*	Nick Jones lab
NJ4	*h+ lag1::KanMX his7-366, ade6-M210*	Nick Jones lab
NJ5	*h- lac1::KanMX his7-366, ade6-M210*	Nick Jones lab
FYL2	*h- ade6-M210 leu1-32 ura4-D18*	This study
FYL19	*h+ lag1-GFP:KanMX lac1::KanMX leu1-32 ura4-D18*	This study
FYL27	*h+ lag1::HphMX his7-366, ade6-M210*	This study
FYL42	*h- lag1-3xHA:KanMX ade6-M210 leu1-32 ura4-D18*	This study
FYL46	*h- lac1::KanMX leu1-32 ura4-D18*	This study
FYL48	*h- lag1::NatMX leu1-32 ura4-D18*	This study
FYL51	*h+ KanMX:nmt1-GFP-lac1 leu1-32 ura4-D18*	This study
FYL52	*h- lag1-GFP:KanMX* *pBip1-mCherry-AHDL:Leu1 ade6-M210 leu1-32 ura4-D18*	This study
FYL65	*h- KanMX:nmt1-GFP-lac1* *lag1-mTomato:NatMX ade6-M210 leu1-32 ura4-D18*	This study
FYL67	*h- KanMX:nmt1-GFP-lac1 pBip1-mCherry-AHDL:Leu1 ade6-M210 leu1-32 ura4-D18*	This study
FYL69	*h- KanMX:nmt1-GFP-lac1* *lag1-3xHA:KanMX leu1-32 ura4-D18*	This study
FYL71	*h- KanMX:nmt1-GFP-lac1 lag1::HphMX leu1-32 ura4-D18*	This study

## Data Availability

The data that support the findings of this study are available from the corresponding authors upon reasonable request.

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
