# Peer review of "The Ceramide Synthase Subunit Lac1 Regulates Cell Growth and Size in Fission Yeast"

_ijms, 2021, doi:10.3390/ijms23010303_

Round 1

Reviewer 1 Report

The present manuscript is clear and presented in a well-structured manner.  the way it’s explained all the experimental plan makes me very easier to understand the biological question and the way it has been answered.

I also appreciate the tecnical pipeline they use to approach the topic which is absolutely appropriate.

The cited references are appropriate for the text of the manuscript.

I found the paper relevant for the field and I think it can be published with some revisions.

Figure 2: give specifications about which is the control in the caption and give more explainations about experimental conditions and settings of in vivo imaging in the text of the manuscript.

Figure 3: A) Give more informations in the text and in the caption about which is the immunoblotting and what is represented in each lane  and which is the ladder and then the expected size. Then specify which is the immunoprecipitation.

In particular I found very few details about immunoblotting, even in Materials and Methods. You must indicate the informations I wrote about figure 2 and which kind of antibodies did you use.

Reviewer 2 Report

This work is trying to understand the underlying mechanism of growth control in fission yeast. The paper described the localization of Lac1 and Lag1 and their function in fission yeast. They found that, unlike budding yeast, Lac1, but not Lag1, is required for proper control of cell growth and size in fission yeast. The paper is interesting, however, the figures in this manuscript are poorly displayed. All labels are missing. So, it is really difficult to do understand the paper and do a full evaluation of this paper. Please find the requested additional experiments and textual changes:
1 The introduction provides good background about the growth mechanisms in the budding yeast. However, the paper is focused on the growth pathways in the fission yeast. So, the difference between the budding yeast and fission yeast should be highlighted. Otherwise, the biological significance of this work is not clear. The introduction does not provide a good rationale why authors test this in this model. 
2 All figures for some reason are black background. The label for the individual panels and the label for experimental groups are missing. I don’t know which figure is figure A or B or C…    It is hard to really understand all the figures. 
3 The sentence “Fission yeast Lac1 shares 37.7% of identical 75 residues with LAC1 from budding yeast, compared to 30.3% with fission yeast Lag1” is confusing. Please make it clear. 
4 Due to the missing label, it is hard to interpret this data.  The paper claims that Lac1 and Lag1 localized at the endoplasmic reticulum. However, this conclusion is from the ectopic expression data. It would be better if you could show the endogenous Lac1 and Lag1 localization. This result should be verified by using expression cassettes that are integrated into the genome. This prevents artifacts from overexpression. Same thing for Figure3, can authors provide evidence that the localization of endogenous expressed Lac1 or Lag1 would not be affected by Lag1∆ or lac1∆? 
5 It is hard to know if the experiments in figure4 have proper control due to the figure problem. If the results are as claimed, it is very interesting. Lag1 single mutants had no effect on cell size at division, can the author try to explain why lag1 and lac1 double mutants were inviable in normal conditions?

Round 2

Reviewer 1 Report

I continue to find this paper relevant for the field and intriguing about the aim and the technical approach. It gives also a pertinent end axhaustive overview of yeast biology. I find really important each reference to previous literature and to what is known about budding yeast to fully make clear to the reader the starting point of the work.

However I find some corrections to make.

Actually I find you transposed almost adequately the comments of the reviewers.

There are already some problems about formatting and photo and pages layout (i.e. In figure 1, now I can’t see the phylogenetic tree) .

Figure 3.A and in general all about western blot. Usually you should show the blot photo as the merge of the chemiluminescence’s photo (samples) and the colorimetric one (ladder). Viceversa, you have to indicate the size of the protein bands at the side of the photo since their molecular weight is the main (not absolutely the unique, but it could be considered enough) value we have for evaluating that antibody works specifically .

Reviewer 2 Report

Thank authors for properly answering my previous questions. As I mentioned earlier, the first draft had serious technical problems, which cause huge problems reading the paper. Thank the authors fix the problem of the figure. After carefully reading the paper, I have the following questions.

1 In figure 1 author did multiple alignments for Lac1 and Lag 1 in 2 different types of yeast. The author claimed that "Fission yeast Lac1 shares 37.7% of identical 82 residues with Lac1 from budding yeast. In comparison, fission yeast Lac1 and Lag1shares only 30.3% with fission yeast Lag1 of identical residues (Figure 1B)." I do not think this is the right comparison and am not sure what the authors are trying to conclude here.  You cannot compare the similarity of Lac in two different yeasts and the similarity between Lac1 and Lag1 in one yeast.  Lac1 and Lag1 are different genes. It is normal that they have less similarity. Compare with them, you cannot really conclude that 37.7% similarly is high.  If you are trying to claim that Lac1 is highly conserved in different types of yeast. You should provide an example of a gene that shares less similarity in two different yeasts (less than 37.7%).  For example, what is the similarity of Lag1 in two different yeasts?

Figure1C is not properly displayed.

In fact, the description of the first paragraph in the results is really confusing. The authors are not making their points clear. In my opinion, figure 1 should be included in the supplemental figures.

2 In figure 2, why are there two inserted pictures? The current figure gives the audience a wrong impression that these three yeast cells are in the same field.

3 In figure 3A, can the authors explain why the GFP signal cannot be detected from the total samples (lanes 4 and 6) in the paper? There is no detailed explanation for the result.  The original image of western blotting shows that the anti-HA and anti-GFP results are not detected from the same membrane. Why is that? Please clarify this in the paper. Please provide at least one more time of the replication result in the supplemental result. Please clarify the replication times of the experiment in the figure legend.

Make sure to have clearly explained all labels in the figure legend. For example, low exp, high exp.

Please rewrite these sentences precisely and clearly: Live-cell imaging of exponentially growing cells Lag1-GFP lac1∆ 183 and GFP-Lac1 lag1∆ revealed that localization at the ER was not impaired (Figure 3B)”

“Thus, localization of Lac1/Lag1 at the ER shows no interdependency, which supports previous data claiming that both subunits have independent ceramide synthase activity [8,9].”

4 The author claimed that "Temperature did not increase defects in proliferation. However, you did not provide any direct evidence for this conclusion. Following this sentence, the description: " Similar results were observed when cells were grown in YES liquid media at 25°C (Figure 4B)". “Similar results” means what? The temperature does not affect proliferation?  This is not what the figure is showing.  Please be precise about the interpretation of the result.

Where is the method for cell length measurement? Figure4 C is the quantification data of figure4D, they should be displayed as are one figure. Please show all individual data points in figure 4C and clarify the method of statistical analysis.

The authors claimed that: Notably, the decrease in cell size in lac1∆ cells was partially reduced at 36°C.” No statistical comparison was done among these groups. Please indicate this in figure4C.

5 The writing for the figure5 results (line 269-292) needs to be improved. The description is not precise and concise. Also, make sure the logic is good. For example, we found a low signal band in lag1∆ cells that could correspond to PH-Cer. Then following sentences back to lipid a, b and c. These are really confusing.

 In figure 5, can the authors show individual data points for bar graphs? The authors claimed that "While no significant changes were found in Lag1 deprived cells", please add statistical analysis for these groups in figure5B.

 "lac1∆ cells exhibited major changes in lipid composition”  why do the authors only show the quantification data for IPC? How about other lipids?

Can the authors explain why PH-Cer was not detected in wt S pombe.? In discussion, the author claimed that "In wild-type cells, long and short-chain PH-Cer and DH-Cer are likely converted into IPC and MIPC".  But for lac1∆, the authors claimed that "the lack of PH-Cer could be responsible for the defects in cell growth and size found in lac1∆ cells." I don't understand the logic here.  Why lacking PH-Cer in wt is not a problem? How do you know lacking PH-Cer is responsible for the deficiency of lac1∆?

6 It is clear that Lag1 is important for long-chain DH-Cer synthesis. In discussion, "lag1∆ cells are not able to synthesize long-chain DH-Cer" but maintain normal cell size. Does this mean that DH-cer is not required for the regulation of cell size? Please discuss this in the discussion. This is very important because otherwise, it is not convincing that "Our genetic and lipid analysis strongly suggest that Lac1 is the fundamental subunit of the ceramide synthase in fission yeast."   and "The results shown here reinforce our hypothesis that ceramide signaling strongly influences control of cell growth and size in eukaryotic cells".

7 The overall writing can be more precise and concise.
